# ON THE POWER OF ABSTENTION AND DATA-DRIVEN DECISION MAKING FOR ADVERSARIAL ROBUSTNESS

## ABSTRACT

We formally define a feature-space attack where the adversary can perturb data-points by arbitrary amounts but in restricted directions. By restricting the attack to a small random subspace, our model provides a clean abstraction for non-Lipschitz networks which map small input movements to large feature movements. We prove that classifiers with the ability to abstain are provably more powerful than those that cannot in this setting. Specifically, we show that no matter how well-behaved the natural data is, any classifier that cannot abstain will be defeated by such an adversary. However, by allowing abstention, we give a parameterized algorithm with provably good performance against such an adversary when classes are reasonably well-separated in feature space and the dimension of the feature space is high. We further use a data-driven method to set our algorithm parameters to optimize over the accuracy vs. abstention trade-off with strong theoretical guarantees. Our theory has direct applications to the technique of contrastive learning, where we empirically demonstrate the ability of our algorithms to obtain high robust accuracy with only small amounts of abstention in both supervised and self-supervised settings. Our results provide a first formal abstention-based gap, and a first provable optimization for the induced trade-off in an adversarial defense setting.

## 1 INTRODUCTION

A substantial body of work has shown that deep networks can be highly susceptible to adversarial attacks, in which minor changes to the input lead to incorrect, even bizarre classifications (Nguyen et al., 2015; Moosavi-Dezfooli et al., 2016; Su et al., 2019; Brendel et al., 2018; Shamir et al., 2019). Much of this work has considered $\ell_p$-norm adversarial examples, but there has also been recent interest in exploring adversarial models beyond bounded $\ell_p$-norm (Brown et al., 2018; Engstrom et al., 2017; Gilmer et al., 2018; Xiao et al., 2018; Alaifari et al., 2019). What these results have in common is that changes that either are imperceptible or should be irrelevant to the classification task can lead to drastically different network behavior.

One reason for this vulnerability to adversarial attack is the non-Lipschitzness property of typical neural networks: small but adversarial movements in the input space can often produce large perturbations in the feature space. In this work, we consider the question of whether non-Lipschitz networks are intrinsically vulnerable, or if they could still be made robust to adversarial attack, in an abstract but (we believe) instructive adversarial model. In particular, suppose an adversary, by making an imperceptible change to an input $x$, can cause its representation $F(x)$ in feature space (the penultimate layer of the network) to move by an arbitrary amount: will such an adversary always win? Clearly if the adversary can modify $F(x)$ by an arbitrary amount in an arbitrary direction, then yes. But what if the adversary can modify $F(x)$ by an arbitrary amount but only in a *random* direction (which it cannot control)? In this case, we show an interesting dichotomy: if the classifier must output a classification on any input it is given, then yes the adversary will still win, no matter how well-separated the classes are in feature space and no matter what decision surface the classifier uses. However, if the classifier is allowed to abstain, then it can defeat such an adversary so long as natural data of different classes are reasonably well-separated in feature space. Our results hold for generalizations of these models as well, such as adversaries that can modify feature representations in random low-dimensional subspaces, or directions that are not completely random. More broadly, our results provide a theoretical explanation for the importance of allowing abstaining, or selective classification, in the presence of adversarial attack.

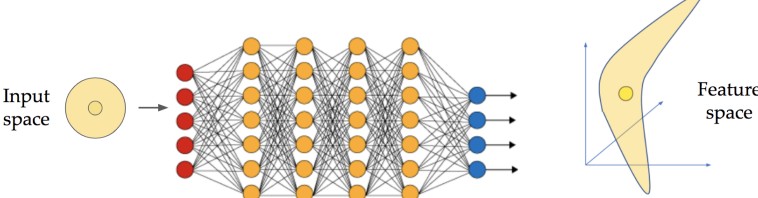

Figure 1: Illustration of a non-Lipschitz feature mapping using a deep network.

Apart from providing a useful abstraction for non-Lipschitz feature embeddings, our model may be viewed as capturing an interesting class of real attacks. There are various global properties of an image, such as brightness, contrast, or rotation angle whose change might be "perceptible but not relevant" to classification tasks. Our model could also be viewed as an abstraction of attacks of that nature. Feature space attacks of other forms, where one can perturb abstract features denoting styles, including interpretable styles such as vivid colors and sharp outlines and uninterpretable ones, have also been empirically studied in (Xu et al., 2020; Ganeshan & Babu, 2019).

An interesting property of our model is that it is critical to be able to refuse to predict: any algorithm which always predicts a class label—therefore without an ability to abstain—is guaranteed to perform poorly. This provides a first formal hardness result about abstention in adversarial defense, and also a first provable negative result in feature-space attacks. We therefore allow the algorithm to output "don't know" for some examples, which, as a by-product of our algorithm, serves as a detection mechanism for adversarial examples. It also results in an interesting trade-off between robustness and accuracy: by controlling how frequently we refuse to predict, we are able to trade (robust) precision off against recall. We also provide results for how to provably optimize for such a trade-off using a data-driven algorithm. Our strong theoretical advances are backed by empirical evidence in the context of contrastive learning (He et al., 2020; Chen et al., 2020; Khosla et al., 2020).

### 1.1 OUR CONTRIBUTIONS

Our work tackles the problem of defending against adversarial perturbations in a *random feature subspace*, and advances the theory and practice of robust machine learning in multiple ways.

- We introduce a formal model that captures feature-space attacks and the effect of non-Lipschitzness of deep networks which can magnify input perturbations.
- We begin our analysis with a hardness result concerning defending against adversary without the option of "don't know". We show that *all* classifiers that partition the feature space into two or more classes—thus without an ability to abstain—are provably vulnerable to adversarial examples for at least one class of examples with nearly half probability.
- We explore the power of abstention option: a variant of nearest-neighbor classifier with the ability to abstain is provably robust against adversarial attacks, even in the presence of outliers in the training data set. We characterize the conditions under which the algorithm does not output "don't know" too often.
- We leverage and extend dispersion techniques from data-driven decision making, and present a novel data-driven method for learning data-specific optimal hyperparameters in our defense algorithms to simultaneously obtain high robust accuracy and low abstention rates. Unlike typical hyperparameter tuning, our approach provably converges to a global optimum.
- Experimentally, we show that our proposed algorithm achieves *certified* adversarial robustness on representations learned by supervised and self-supervised contrastive learning. Our method significantly outperforms algorithms without the ability to abstain.

## 2 RELATED WORK

**Adversarial robustness with abstention options.** Classification with abstention option (a.k.a. selective classification (Geifman & El-Yaniv, 2017)) is a relatively less explored direction in the adversarial machine learning. Hosseini et al. (2017) augmented the output class set with a NULL label and trained the classifier to reject the adversarial examples by classifying them as NULL; Stutz et al. (2020) and Laidlaw & Feizi (2019) obtained robustness by rejecting low-confidence adversarial examples

according to confidence thresholding or predictions on the perturbations of adversarial examples. Another related line of research to our method is the detection of adversarial examples (Grosse et al., 2017; Li & Li, 2017; Carlini & Wagner, 2017; Ma et al., 2018; Meng & Chen, 2017; Metzen et al., 2017; Bhagoji et al., 2018; Xu et al., 2017; Hu et al., 2019). However, theoretical understanding behind the empirical success of adversarial defenses with an abstention option remains elusive.

**Data-driven decision making.** Data-driven algorithm selection refers to choosing a good algorithm from a parameterized family of algorithms for given data. It is known as "hyperparameter tuning" to machine learning practitioners and typically involves a "grid search", "random search" (Bergstra & Bengio (2012)) or gradient-based search, with no guarantees of convergence to a global optimum. It was formally introduced to the theory of computing community by Gupta & Roughgarden (2017) as a learning paradigm, and was further extended in (Balcan et al., 2017). The key idea is to model the problem of identifying a good algorithm from data as a statistical learning problem. The technique has found useful application in providing provably better algorithms for several domains including clustering, mechanism design, and mixed integer programs, and providing guarantees like differential privacy and adaptive online learning (Balcan et al., 2018a;b; 2020). For learning in an adversarial setting, we provide the first demonstration of the effectiveness of data-driven algorithm selection in a defense method to optimize over the accuracy-abstention trade-off with strong theoretical guarantees.

## 3 PRELIMINARIES

**Notation.** We will use *bold lower-case* letters such as $\boldsymbol{x}$ and $\boldsymbol{y}$ to represent vectors, *lower-case* letters such as $x$ and $y$ to represent scalars, and *calligraphy capital* letters such as $\mathcal{X}$, $\mathcal{Y}$ and $\mathcal{D}$ to represent distributions. Specifically, we denote by $\boldsymbol{x} \in \mathcal{X}$ the sample instance, and by $y \in \mathcal{Y}$ the label, where $\mathcal{X} \subseteq \mathbb{R}^{n_1}$ and $\mathcal{Y}$ indicate the image and label spaces, respectively. Denote by $F : \mathcal{X} \to \mathbb{R}^{n_2}$ the *feature embedding* which maps an instance to a high-dimensional vector in the latent space $F(\mathcal{X})$. It can be parameterized, e.g., by deep neural networks. We will frequently use $\mathbf{v} \in \mathbb{R}^{n_2}$ to represent an adversarial perturbation in the feature space. Denote by $\text{dist}(\cdot, \cdot)$ the distance between any two vectors in the image or feature space. Examples of distances include $\text{dist}(\boldsymbol{x}_1, \boldsymbol{x}_2) = \|\boldsymbol{x}_1 - \boldsymbol{x}_2\|$—the one induced by vector norm. We use $\mathbb{B}(\boldsymbol{x}, \tau)$ to represent a neighborhood of $\boldsymbol{x}$: $\{\boldsymbol{x}' : \text{dist}(\boldsymbol{x}, \boldsymbol{x}') \leq \tau\}$ in the image or feature space. We will frequently denote by $\mathcal{D}_{\mathcal{X}}$ the distribution of instances in the input space, by $\mathcal{D}_{\mathcal{X}|y}$ the distribution of instances in the input space conditioned on the class $y$, by $\mathcal{D}_{F(\mathcal{X})}$ the distribution of features, and by $\mathcal{D}_{F(\mathcal{X})|y}$ the distribution of features conditioned on the class $y$.

### 3.1 RANDOM FEATURE SUBSPACE THREAT MODEL

In principle, the adversarial example for a given labeled data $(\boldsymbol{x}, y)$ is a data point $\boldsymbol{x}'$ that causes a classifier to output a different label on $\boldsymbol{x}'$ than the true label $y$. Probably one of the most popular adversarial examples is the norm-bounded perturbation in the input space. Despite a large literature devoted to defending against norm-bounded adversary by improving the Lipschitzness of neural network as a function mapping from input space to feature space (Zhang et al., 2019; Yang et al., 2020), it is typically not true that small perturbation in the input space necessarily implies small modification in the feature space. In this paper, we study a threat model where an adversary can modify the data by a large amount in the feature space. Note that because this large modification in feature space is assumed to come from a small perturbation in input space, we always assume that the *true correct label $y$ is the same for $x'$ as for $x$*. Our model highlights the power of abstention in the adversarial learning: there is a provable separation when we have and do not have an abstention option under our threat model.

**Our threat model.** In the setting of (robust) representation learning, we are given a set of training instances $\boldsymbol{x}_1, ..., \boldsymbol{x}_m \in \mathcal{X}$. Let $\boldsymbol{x}$ be an $n_1$-dimensional test input for classification. The input is embedded into a high $n_2$-dimensional feature space using a deep neural network $F$. We predict the class of $\boldsymbol{x}$ by a prediction function on $F(\boldsymbol{x})$ which can potentially output "don't know". The adversary may corrupt $F(\boldsymbol{x})$ such that the modified feature vector is restricted in a random $n_3$-dimensional affine subspace denoted by $\mathcal{S} + \{F(\boldsymbol{x})\}$, while the perturbation magnitude might be arbitrarily large. The adversary is given access to everything including $F$, $\boldsymbol{x}$, $\mathcal{S}$ and the true label of $\boldsymbol{x}$. Throughout the paper, we will refer *adversary* and *adversarial example* to this threat model.

---

**Algorithm 1** ROBUSTCLASSIFIER$(\tau, \sigma)$

---

1: **Input:** A test feature $F(\boldsymbol{x})$ (potentially an adversarial example), a set of training features $F(\boldsymbol{x}_i)$ and their labels $y_i$, $i \in [m]$, a threshold parameter $\tau$, a separation parameter $\sigma$.
2: **Preprocessing:** Delete training examples $F(\boldsymbol{x}_i)$ if $\min_{j \in [m], y_i \neq y_j} \mathsf{dist}(F(\boldsymbol{x}_i), F(\boldsymbol{x}_j)) < \sigma$
3: **Output:** A predicted label of $F(\boldsymbol{x})$, or "don't know".
4: **if** $\min_{i \in [m]} \mathsf{dist}(F(\boldsymbol{x}), F(\boldsymbol{x}_i)) < \tau$ **then**
5:     Return $y_{\arg\min_{i \in [m]} \mathsf{dist}(F(\boldsymbol{x}), F(\boldsymbol{x}_i))}$
6: **else**
7:     Return "don't know"

---

## 3.2 A META-ALGORITHM FOR INFERENCE-TIME ROBUSTNESS

Given a test data $\boldsymbol{x}$, let $r$ denote the shortest distance between $F(\boldsymbol{x})$ and any training embedding $F(\boldsymbol{x}_i)$ of different labels. Throughout the paper, we consider the prediction rule that we classify an unseen (and potentially adversarially modified) example with the class of its nearest training example provided that the distance between them is at most $\tau$; otherwise the algorithm outputs "don't know" (see Algorithm 1 and Figure 2). The adversary is able to corrupt $F(\boldsymbol{x})$ by a carefully-crafted perturbation along a random direction, i.e., $F(\boldsymbol{x}) + \mathbf{v}$, where $\mathbf{v}$ is an adversarial vector of arbitrary length in a random $n_3$-dimensional subspace of $\mathbb{R}^{n_2}$. The parameter $\tau$ trades the success rate off against the abstention rate; when $\tau \to \infty$, our algorithm is equivalent to the nearest-neighbor algorithm. We also preprocess to remove outliers and points too close to them.

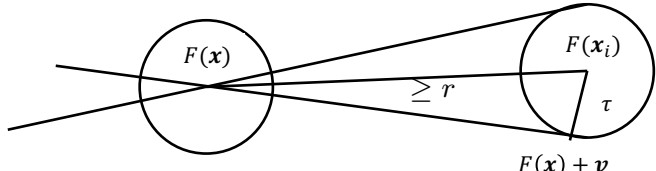

Figure 2: Adversarial misclassification for nearest-neighbor predictor.

## 4 NEGATIVE RESULTS WITHOUT AN ABILITY TO ABSTAIN

Several negative results are known for defending against adversarial examples beyond norm-bounded settings. For example, Shamir et al. (2019) provably show existence of targeted adversarial examples with small hamming distance in the input space to their clean examples. For feature-space attacks, several empirical negative results are known (Xu et al., 2020; Ganeshan & Babu, 2019). We present a hardness result concerning defenses without an ability to abstain, and prove that such defenses are inevitably doomed against our feature-space attacks.

**Theorem 4.1.** *For any classifier that partitions $\mathbb{R}^{n_2}$ into two or more classes, any data distribution $\mathcal{D}$, any $\delta > 0$ and any feature embedding $F$, there must exist at least one class $y^*$, such that for at least a $1 - \delta$ probability mass of examples $\boldsymbol{x}$ from class $y^*$ (i.e., $\boldsymbol{x}$ is drawn from $\mathcal{D}_{\mathcal{X}|y^*}$), for a random unit-length vector $\mathbf{v}$, with probability at least $1/2 - \delta$ for some $\delta_0 > 0$, $F(\boldsymbol{x}) + \delta_0 \mathbf{v}$ is not labeled $y^*$ by the classifier. In other words, there must be at least one class $y^*$ such that for at least $1 - \delta$ probability mass of points $\boldsymbol{x}$ of class $y^*$, the adversary wins with probability at least $1/2 - \delta$.*

*Proof.* Without loss of generality, we assume that the feature embedding $F$ is an identity mapping. Define $r_\delta$ to be a radius such that for every class $y$, at least a $1 - \delta$ probability mass of examples $\boldsymbol{x}$ of class $y$ lie within distance $r_\delta$ of the origin. Let $R = r_\delta \sqrt{n_2}/\delta$. $R$ is defined to be large enough such that if we take a ball of radius $R$ and move it by a distance $r_\delta$, at least a $1 - \delta$ fraction of the volume of the new ball is inside the intersection with the old ball. Now, let $\mathcal{B}$ be the ball of radius $R$ centered at the origin. Let $\mathsf{vol}(\mathcal{B})$ denote the volume of $\mathcal{B}$ and let $\mathsf{vol}_y(\mathcal{B})$ denote the volume of the subset of $\mathcal{B}$ that is assigned label $y$ by the classifier. Let $y^*$ be any label such that $\mathsf{vol}_{y^*}(\mathcal{B})/\mathsf{vol}(\mathcal{B}) \leq 1/2$. Such a class $y^*$ exists because we do not have the option to output "don't know". Now by the definition of $y^*$, a point $\mathbf{z}$ picked uniformly at random from $\mathcal{B}$ has probability at least $1/2$ of being classified differently from $y^*$. This implies that, by the definition of $R$, if $\boldsymbol{x}$ is within distance $r_\delta$ of the origin,

then a point $\mathbf{z}_x$ that is picked uniformly at random in the ball $\mathcal{B}_x$ of radius $R$ centered at $\boldsymbol{x}$ has probability at least $1/2 - \delta$ of being classified differently from $y^*$. This immediately implies that if we choose a random unit-length vector $\mathbf{v}$, then with probability at least $1/2 - \delta$, there exists $\delta_0 > 0$ such that $\boldsymbol{x} + \delta_0 \mathbf{v}$ is classified differently from $y^*$, since we can think of choosing $\mathbf{v}$ by first sampling $\mathbf{z}_x$ from $\mathcal{B}_x$ and then defining $\mathbf{v} = (\mathbf{z}_x - \boldsymbol{x})/\|\mathbf{z}_x - \boldsymbol{x}\|_2$. So, the theorem follows from the fact that, by the definition of $r_\delta$, at least $1 - \delta$ probability mass of examples $\boldsymbol{x}$ from class $y^*$ are within distance $r_\delta$ of the origin. $\qquad\square$

We remark that our lower bound applies to any classifier and exploits the fact that a classifier without abstention must label the entire feature space. For a simple linear decision boundary (center of Figure 3), a perturbation in any direction (except parallel to the boundary) can cross the boundary with an appropriate magnitude. The left and right figures show that if we try to 'bend' the decision boundary to 'protect' one of the classes, the other class is still vulnerable. Our argument formalizes and generalizes this intuition, and shows that there must be at least one vulnerable class irrespective of how you may try to shape the class boundaries, where the adversary succeeds in a large fraction of directions.

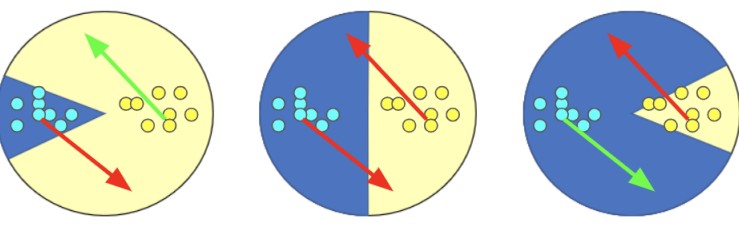

Figure 3: A simple example to illustrate Theorem 4.1.

Theorem 4.1 implies that *all* classifiers that partitions $\mathbb{R}^{n_2}$ into two or more classes—thus without an ability to abstain—are vulnerable to adversarial examples for at least one class of data with nearly half probability. Despite much effort has been devoted to empirically investigating the power of "don't know" in the adversarial robustness, theoretical understanding behind the empirical success of these methods remains elusive. To the best of our knowledge, our work is the first result that provably demonstrates the power of "don't know" in the algorithmic design of adversarially robust classifiers.

## 5 POSITIVE RESULTS WITH AN ABILITY TO ABSTAIN

Theorem 4.1 gives a hardness result of robust classification without abtention. In this section, we explore the power of abstaining and show classifiers with an ability to abstain are provably robust.

Given a test instance $\boldsymbol{x} \sim \mathcal{D}_\mathcal{X}$, recall that $r$ denotes the shortest distance between $F(\boldsymbol{x}) \in \mathbb{R}^{n_2}$ and any training embedding $F(\boldsymbol{x}_i) \in \mathbb{R}^{n_2}$ with a different label. The adversary is allowed to corrupt $F(\boldsymbol{x})$ with an arbitrarily large perturbation in a uniform-distributed subspace $S$ of dimension $n_3$. Consider the prediction rule that we classify the unseen example $F(\boldsymbol{x}) \in \mathbb{R}^{n_2}$ with the class of its nearest training example provided that the distance between them is at most $\tau$; otherwise the algorithm outputs "don't know" (see Algorithm 1 when $\sigma = 0$). Denote by $\mathcal{E}_{\text{adv}}^{\boldsymbol{x}}(f) := \mathbb{E}_{S \sim \mathcal{S}} \mathbf{1}\{\exists \mathbf{e} \in S + F(\boldsymbol{x}) \subseteq \mathbb{R}^{n_2} \text{ s.t. } f(\mathbf{e}) \neq \boldsymbol{y} \text{ and } f(\mathbf{e}) \text{ does not abstain}\}$ the robust error of a given classifier $f$ for classifying instance $\boldsymbol{x}$. Our analysis leads to the following positive results on this algorithm.

**Theorem 5.1.** *Let $\boldsymbol{x} \sim \mathcal{D}_\mathcal{X}$ be a test instance, $m$ be the number of training examples and $r$ be the shortest distance between $F(\boldsymbol{x})$ and $F(\boldsymbol{x}_i)$ where $\boldsymbol{x}_i$ is a training point from a different class. Suppose $\tau = o\left(r\sqrt{1 - \frac{n_3}{n_2}}\right)$. The robust error of Algorithm 1, $\mathcal{E}_{\text{adv}}^{\boldsymbol{x}}(\text{ROBUSTCLASSIFIER}(\tau, 0))$, is*

*at most* $m\left(\dfrac{c\tau}{r\sqrt{1 - \frac{n_3}{n_2}}}\right)^{n_2 - n_3} + m c_0^{n_2 - n_3}$, *where $c > 0$ and $0 < c_0 < 1$ are absolute constants.*

*Proof Sketch.* We begin our analysis with the case of $n_3 = 1$. Suppose we have a training example $\boldsymbol{x}'$ of another class, and suppose $F(\boldsymbol{x})$ and $F(\boldsymbol{x}')$ are at distance $D$ in the feature space. Because $\tau = o(D)$, the probability that the adversary can move $F(\boldsymbol{x})$ to within distance $\tau$ of $F(\boldsymbol{x}')$ should

be roughly the ratio of the surface area of a sphere of radius $\tau$ to the surface area of a sphere of radius $D$, which is at most $\left(\mathcal{O}\left(\frac{\tau}{D}\right)\right)^{n_2-1} \leq \left(\mathcal{O}\left(\frac{\tau}{r}\right)\right)^{n_2-1}$. The analysis for the general case of $n_3$ follows from a pealing argument: note that the random subspace in which the adversary vector is restricted to lie can be constructed by first sampling a vector $\mathbf{v}_1$ uniformly at random from a unit sphere in the ambient space $\mathbb{R}^{n_2}$ centered at 0; fixing $\mathbf{v}_1$, we then sample a vector $\mathbf{v}_2$ uniformly at random from a unit sphere in the null space of $\mathsf{span}\{\mathbf{v}_1\}$; we repeat this procedure $n_3$ times and let $\mathsf{span}\{\mathbf{v}_1, \mathbf{v}_2, ..., \mathbf{v}_{n_3}\}$ be the desired adversarial subspace. For each step of construction, we apply the same argument as that of $n_3 = 1$ with $D = \Omega\left(r\sqrt{\frac{n_2-i}{n_2}}\right)$ by a high probability, if we project $F(\boldsymbol{x})$ and $F(\boldsymbol{x}')$ to a random subspace of dimension $n_2 - i$. Finally, a union bound over $m$ training points completes the proof. $\qquad\square$

**Trade-off between success probability and abstention rate.** Theorem 5.1 captures the trade-off between the success probability of an algorithm and the abstention rate: a smaller value of $\tau$ increases the success probability of the algorithm, while it also encourages Algorithm 1 to output "don't know" more often. A related line of research to this observation is the trade-off between robustness and accuracy: Zhang et al. (2019); Tsipras et al. (2019) showed that there might be no predictor in the hypothesis class that has low natural and robust errors; even such a predictor exists for the well-separated data (Yang et al., 2020), Raghunathan et al. (2020) showed that the natural error could increase by adversarial training if we only have finite number of data. To connect the two trade-offs, we note that a high success probability of ROBUSTCLASSIFIER$(\tau, 0)$ in Algorithm 1 tends to avoid the algorithm from predicting wrong labels for adversarial examples, while the associated high abstention rate encourages the algorithm to output "don't know" even for natural examples, thus leading to a trivial non-accurate classifier.

## 5.1 A MORE GENERAL ADVERSARY WITH BOUNDED DENSITY

We extend our results to a more general class of adversaries, which have a bounded distribution over the space of linear subspaces of a fixed dimension $n_3$ and the adversary can perturb a test feature vector arbitrarily in the sampled adversarial subspace.

**Theorem 5.2.** *Consider the setting of Theorem 5.1, with an adversary having a $\kappa$-bounded distribution over the space of linear subspaces of a fixed dimension $n_3$ for perturbing the test point. If $\mathbf{E}(\tau, r)$ denotes the bound on error rate in Theorem 5.1 for ROBUSTCLASSIFIER$(\tau, 0)$ in Algorithm 1, then the error bound of the same algorithm against the $\kappa$-bounded adversary is $\mathcal{O}(\kappa\mathbf{E}(\tau, r))$.*

## 5.2 OUTLIER REMOVAL AND IMPROVED UPPER BOUND

The upper bounds above assume that the data is well-separated in the feature space. For noisy data and good-but-not-perfect embeddings, the condition may not hold. In Theorem E.1 (in Appendix E) we show that we obtain almost the same upper bound on failure probability under weaker assumptions by exploiting the noise removal threshold $\sigma$.

## 5.3 CONTROLLING ABSTENTION RATE ON NATURAL DATA

We show that we can control the frequency of outputting "don't know", when the data are nicely distributed according to the following generative assumption. Intuitively, it says that for every label class one can cover most of the distribution of the class with (potentially overlapping) balls of a fixed radius, each having a small lower bound on the density contained. This holds for well-clustered datasets (as is typical for feature data) for a sufficiently large radius.

**Assumption 1.** *We assume that at least $1 - \delta$ fraction of mass of the marginal distribution $\mathcal{D}_{F(\mathcal{X})|y}$ over $\mathbb{R}^{n_2}$ can be covered by $N$ balls $\mathbb{B}_1$, $\mathbb{B}_2$, ... $\mathbb{B}_N$ of radius $\tau/2$ and of mass $\mathrm{Pr}_{\mathcal{D}_{F(\mathcal{X})}}[\mathbb{B}_k] \geq \frac{C_0}{m}\left(n_2 \log m + \log \frac{4N}{\beta}\right)$, where $C_0 > 0$ is an absolute constant and $\delta, \beta \in (0, 1)$.*

Our analysis leads to the following guarantee on the abstention rate.

**Theorem 5.3.** *Suppose that $F(\boldsymbol{x}_1), ..., F(\boldsymbol{x}_m)$ are $m$ training instances i.i.d. sampled from marginal distribution $\mathcal{D}_{F(\mathcal{X})}$. Under Assumption 1, with probability at least $1 - \beta/4$ over the sampling, we have $\mathrm{Pr}(\cup_{i=1}^m \mathbb{B}(F(\boldsymbol{x}_i), \tau)) \geq 1 - \delta$.*

Theorem 5.3 implies that when $\Pr[\mathbb{B}_k] \geq \frac{\beta}{N}$ and $m = \Omega(\frac{n_2 N}{\beta} \log \frac{n_2 N}{\beta})$, with probability at least $1 - \beta/4$ over the sampling, we have $\Pr(\cup_{i=1}^m \mathbb{B}(F(\boldsymbol{x}_i), \tau)) \geq 1 - \delta$. Therefore, with high probability, the algorithm will output "don't know" only for an $\delta$ fraction of natural data.

## 6  LEARNING DATA-SPECIFIC OPTIMAL THRESHOLDS

Given an embedding function $F$ and a classifier $f_\tau$ which outputs either a predicted class if the nearest neighbor is within distance $\tau$ of a test point or abstains from predicting, we want to evaluate the performance of $f_\tau$ on a test set $\mathcal{T}$ against an adversary which can perturb a test feature vector in a random subspace $S \sim \mathcal{S}$. To this end, we define $\mathcal{E}_{\mathrm{adv}}(\tau) := \mathbb{E}_{S \sim \mathcal{S}} \frac{1}{|\mathcal{T}|} \sum_{(\boldsymbol{x}, \boldsymbol{y}) \in \mathcal{T}} \mathbf{1}\{\exists \mathbf{e} \in S + F(\boldsymbol{x}) \subseteq \mathbb{R}^{n_2}$ s.t. $f(\mathbf{e}) \neq \boldsymbol{y}$ and $f_\tau(\mathbf{e})$ does not abstain$\}$ as the robust error on the test set $\mathcal{T}$, and $\mathcal{D}_{\mathrm{nat}}(\tau) := \frac{1}{|\mathcal{T}|} \sum_{(\boldsymbol{x}, \boldsymbol{y}) \in \mathcal{T}} \mathbf{1}\{f_\tau(F(\boldsymbol{x})) \text{ abstains}\}$ as the abstention rate on the natural data. $\mathcal{E}_{\mathrm{adv}}(\tau)$ and $\mathcal{D}_{\mathrm{nat}}(\tau)$ are monotonic in $\tau$. The robust error $\mathcal{E}_{\mathrm{adv}}(\tau)$ is optimal at $\tau = 0$, while we abstain from prediction all the time (i.e., $\mathcal{D}_{\mathrm{nat}}(\tau) = 1$). A simple approach is to fix an upper limit $d^*$ on $\mathcal{D}_{\mathrm{nat}}(\tau)$, which corresponds to the maximum abstention rate on natural data under our budget. Then it is straightforward to search for the optimal $\tau^*$ such that $\mathcal{D}_{\mathrm{nat}}(\tau^*) \approx d^*$ by using nearest neighbor distances of test points. For $\tau < \tau^*$ we have a higher abstention rate, and when $\tau > \tau^*$ we have a higher robust error rate. A potential problem with this approach is that $\mathcal{D}_{\mathrm{nat}}(\tau)$ is non-Lipschitz, so small variation in $\tau$ can possibly make the abstention rate significantly higher than $d^*$.

An alternative objective which captures the trade-off between abstention rate and accuracy is defined as $g(\tau) := \mathcal{E}_{\mathrm{adv}}(\tau) + c\mathcal{D}_{\mathrm{nat}}(\tau)$, where $c$ is a positive constant. If, for example, we are willing to take a one percent increase of the abstention rate for a two percent drop in the error rate, we could set $c$ to be $\frac{1}{2}$. We can optimize $g(\tau)$ in a data-driven fashion and obtain theoretical guarantee on the convergence to a global optimum. In the following, we consider the case where the test examples appear in an online fashion in small batches of size $b$, and we set the threshold $\tau$ adaptively by a low-regret algorithm. We note in Corollary 6.3, using online-to-batch conversion, that our results imply a uniform convergence bound for objective $g(\tau)$ in the supervised setting. Details of proofs in this section can be found in Appendix H.

The significance of data-driven design in this setting is underlined by the following two observations. Firstly, as noted above, optimization for $\tau$ is difficult due to the non-Lipschitzness nature of $\mathcal{D}_{\mathrm{nat}}(\tau)$ and the intractability of characterizing the objective function $g(\tau)$ exactly due to $\mathcal{E}_{\mathrm{adv}}(\tau)$. Secondly, the optimal value of $\tau$ can be a complex function of the data geometry and sampling rate. We illustrate this by exact computation of optimal $\tau$ for a simple intuitive setting: consider a binary classification problem where the features lie uniformly on two one-dimensional manifolds embedded in two-dimensions (i.e., $n_2 = 2$, see Figure 4). Assume that the adversary perturbs in a uniformly random direction ($n_3 = 1$). For this setting, in Appendix J we show that

**Theorem 6.1.** *Let* $\tau^* := \arg\max_{\tau \in \mathbb{R}^+} g(\tau)$ *and* $\beta = \frac{2\pi cr}{D}$. *For the setting considered above, if we further assume* $D = o(r)$ *and* $m = \omega(\log \beta)$, *then there is a unique value of* $\tau^*$ *in* $[0, D/2)$. *Furthermore, we have* $\tau^* = \Theta\left(\frac{D \log(\beta m)}{m}\right)$ *if* $m > \frac{1}{\beta}$; *otherwise,* $\tau^* = 0$.

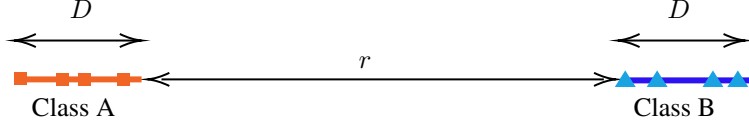

Figure 4: A simple intuitive example where we compute the optimal value of the abstention threshold exactly. Classes A and B are both distributed uniformly on one-dimensional segments of length $D$, embedded collinear and at distance $r$ in $\mathbb{R}^2$.

The remaining section summarizes our main theoretical results.

**Theorem 6.2.** *Assume* $\tau$ *is* $o\left(\min\{m^{-1/n_2}, r\}\right)$, *and the data distribution is continuous,* $\kappa$-*bounded, positive and has bounded partial derivatives. If* $\tau$ *is set using a continuous version of the multiplicative updates algorithm (Algorithm 2 in Appendix H, Balcan et al. (2018a)), then with probability at least* $1 - \delta$, *the expected regret in* $T$ *rounds is bounded by* $O\left(\sqrt{n_2 T \log\left(\frac{\kappa RTmb}{\delta r^{n_2 - n_3}}\right)}\right)$, *where* $R$ *is a bound*

*on the largest distance between any two training points, $b$ is the batch size, and $r$ is the smallest distance between points of different labels.*

**Corollary 6.3.** *Suppose we run the online algorithm of Theorem 6.2 on a validation set of size $T$, and use a randomized threshold $\hat{\tau}$ on the test set drawn from a uniform distribution over the thresholds $\tau_1, \ldots, \tau_T$ used in online learning. If the threshold which maximizes $g(\tau)$ is $\tau^*$, then with probability greater than $1 - \delta$, we have $|\mathbb{E}[g(\hat{\tau})] - g(\tau^*)| \leq O\left(\sqrt{\frac{n_2}{T} \log\left(\frac{\kappa R T m b}{\delta r^{n_2 - n_3}}\right)}\right)$.*

**Remark 1.** *The results can be generalized to a bounded density adversary (Corollary H.3).*

**Remark 2.** *The above analysis can be extended to the problem of optimizing over $\sigma$ by formulating the objective as function of two parameters, $g(\tau, \sigma) := \mathcal{E}_{\text{adv}}(\tau, \sigma) + c\mathcal{D}_{\text{nat}}(\tau, \sigma)$ within a range $\sigma \in [r, s]$. For fixed $\tau$, both $\mathcal{E}_{\text{adv}}(\tau, \sigma)$ and $\mathcal{D}_{\text{nat}}(\tau, \sigma)$ are piece-wise constant and monotonic. The proof of Lipschitzness of the pieces can be adapted easily to the case of $\sigma \geq r$ (Lemma H.2). Discontinuities in $\mathcal{E}_{\text{adv}}(\tau, \cdot)$ and $\mathcal{D}_{\text{nat}}(\tau, \cdot)$ can be bounded using the upper bound $s$ for $\sigma$ (Lemma H.4). Finally, the number of discontinuities in $g(\tau, \sigma)$ in a ball of radius $w$ can be upper bounded by a product of the number of discontinuities in $g(\tau, \cdot)$ and $g(\cdot, \sigma)$ in intervals of width $w$.*

## 7 EXPERIMENTS ON CONTRASTIVE LEARNING

Theorem 5.1 sheds light on algorithmic designs of robust learning of feature embedding $F$. In order to preserve robustness against adversarial examples regarding a given test point $\boldsymbol{x}$, in the feature space the theorem suggests minimizing $\tau$—the closest distance between $F(\boldsymbol{x})$ and any training feature $F(\boldsymbol{x}_i)$ of the same label, and maximizing $r$—the closest distance between $F(\boldsymbol{x})$ and any training feature $F(\boldsymbol{x}_i)$ of different labels. This is conceptually consistent with the spirit of the nearest-neighbor algorithm, a.k.a. contrastive learning when we replace the *max* operator with the *softmax* operator for differentiable training:

$$\min_F -\frac{1}{m} \sum_{i \in [m]} \log\left(\frac{\sum_{j \in [m], j \neq i, y_i = y_j} e^{-\frac{\|F(\boldsymbol{x}_i) - F(\boldsymbol{x}_j)\|^2}{T}}}{\sum_{k \in [m], k \neq i} e^{-\frac{\|F(\boldsymbol{x}_i) - F(\boldsymbol{x}_k)\|^2}{T}}}\right), \tag{1}$$

where $T > 0$ is the temperature parameter. Loss (1) is also known as the soft-nearest-neighbor loss in the context of supervised learning (Frosst et al., 2019), or the InfoNCE loss in the setting of self-supervised learning (He et al., 2020).

### 7.1 CERTIFIED ADVERSARIAL ROBUSTNESS AGAINST EXACT COMPUTATION OF ATTACKS

We verify the robustness of Algorithm 1 when the representations are learned by contrastive learning. Given a embedding function $F$ and a classifier $f$ which outputs either a predicted class or abstains from predicting, recall that we define the natural and robust errors, respectively, as $\mathcal{E}_{\text{nat}}(f) := \mathbb{E}_{(\boldsymbol{x},\boldsymbol{y})\sim\mathcal{D}}\mathbf{1}\{f(F(\boldsymbol{x})) \neq \boldsymbol{y}$ and $f(F(\boldsymbol{x}))$ does not abstain$\}$, and $\mathcal{E}_{\text{adv}}(f) := \mathbb{E}_{(\boldsymbol{x},\boldsymbol{y})\sim\mathcal{D}, S\sim\mathcal{S}}\mathbf{1}\{\exists \mathbf{e} \in S + F(\boldsymbol{x}) \subseteq \mathbb{R}^{n_2}$ s.t. $f(\mathbf{e}) \neq \boldsymbol{y}$ and $f(\mathbf{e})$ does not abstain$\}$, where $S \sim \mathcal{S}$ is a random adversarial subspace of $\mathbb{R}^{n_2}$ with dimension $n_3$. $\mathcal{D}_{\text{nat}}(f) := \mathbb{E}_{(\boldsymbol{x},\boldsymbol{y})\sim\mathcal{D}}\mathbf{1}\{f(F(\boldsymbol{x}))$ abstains$\}$ is the abstention rate on the natural examples. Note that the robust error is always at least as large as the natural error.

**Self-supervised contrastive learning setup.** Our experimental setup follows that of SimCLR (Chen et al., 2020). We use the ResNet-18 architecture (He et al., 2016) for representation learning with a two-layer projection head of width 128. The dimension of the representations is 512. We set batch size 512, temperature $T = 0.5$, and initial learning rate 0.5 which is followed by cosine learning rate decay. We sequentially apply four simple augmentations: random cropping followed by resize back to the original size, random flipping, random color distortions, and randomly converting image to grayscale with a probability of 0.2. In the linear evaluation protocol, we set batch size 512 and learning rate 1.0 to learn a linear classifier in the feature space by empirical risk minimization.

**Supervised contrastive learning setup.** Our experimental setup follows that of Khosla et al. (2020). We use the ResNet-18 architecture for representation learning with a two-layer projection head of width 128. The dimension of the representations is 512. We set batch size 512, temperature $T = 0.1$, and initial learning rate 0.5 which is followed by cosine learning rate decay. We sequentially apply four simple augmentations: random cropping followed by resize back to the original size, random

Table 1: Natural error $\mathcal{E}_{\text{nat}}$ and robust error $\mathcal{E}_{\text{adv}}$ on the CIFAR-10 dataset when $n_3 = 1$ and the 512-dimensional representations are learned by contrastive learning, where $\mathcal{D}_{\text{nat}}$ represents the fraction of each algorithm's output of "don't know" on the natural data. We report values for $\sigma \approx \tau$ as they tend to give a good abstention-error trade-off w.r.t. $\sigma$.

| Contrastive | | Linear Protocol | | Ours ($\tau = 3.0$) | | | Ours ($\tau = 2.0$) | | |
|---|---|---|---|---|---|---|---|---|---|
| | | $\mathcal{E}_{\text{nat}}$ | $\mathcal{E}_{\text{adv}}$ | $\mathcal{E}_{\text{nat}}$ | $\mathcal{E}_{\text{adv}}$ | $\mathcal{D}_{\text{nat}}$ | $\mathcal{E}_{\text{nat}}$ | $\mathcal{E}_{\text{adv}}$ | $\mathcal{D}_{\text{nat}}$ |
| ($\sigma = 0$) | Self-supervised | 8.9% | 100.0% | 15.4% | 40.7% | 2.2% | 14.3% | 26.2% | 28.7% |
| | Supervised | 5.6% | 100.0% | 5.7% | 60.5% | 0.0% | 5.7% | 33.4% | 0.0% |
| ($\sigma = 0.9\tau$) | Self-supervised | 8.9% | 100.0% | 7.2% | 9.4% | 12.9% | 10.0% | 17.7% | 29.9% |
| | Supervised | 5.6% | 100.0% | 6.2% | 18.9% | 0.0% | 5.6% | 22.0% | 0.1% |
| ($\sigma = \tau$) | Self-supervised | 8.9% | 100.0% | 1.1% | 1.2% | 33.4% | 2.1% | 3.1% | 49.9% |
| | Supervised | 5.6% | 100.0% | 1.9% | 2.8% | 10.6% | 4.1% | 4.8% | 3.3% |

flipping, random color distortions, and randomly converting image to grayscale with a probability of 0.2. In the linear evaluation protocol, we set batch size 512 and learning rate 5.0 to learn a linear classifier in the feature space by empirical risk minimization.

In both self-supervised and supervised setups, we compare the robustness of the linear protocol with that of our defense protocol in Algorithm 1 under exact computation of adversarial examples using a convex optimization program in $n_3$ dimensions and $m$ constraints. Algorithm 4 in the appendix provides an efficient implementation of the attack.

**Experimental results.** We summarize our results in Table 1. Comparing with a linear protocol, our algorithms have much lower robust error. Note that even if abstention is added based on distance from the linear boundary, sufficiently large perturbations will ensure the adversary can always succeed. For an approximate adversary which can be efficiently implemented for large $n_3$, see Appendix L.2.

### 7.2 ROBUSTNESS-ABSTENTION TRADE-OFF

The threshold parameter $\tau$ captures the trade-off between the robust accuracy $\mathcal{A}_{\text{adv}} := 1 - \mathcal{E}_{\text{adv}}$ and the abstention rate $\mathcal{D}_{\text{nat}}$ on the natural data. We report both metrics for different values of $\tau$ for supervised and self-supervised constrastive learning. The supervised setting enjoys higher adversarial accuracy and a smaller abstention rate for fixed $\tau$'s due to the use of extra label information. We plot $\mathcal{A}_{\text{adv}}$ against $\mathcal{D}_{\text{nat}}$ for Algorithm 1 as hyperparameters vary. For small $\tau$, both accuracy and abstention rate approach 1.0. As the threshold increases, the abstention rate decreases rapidly and our algorithm enjoys good accuracy even with small abstention rates. For $\tau \to \infty$ (i.e. the nearest neighbor search), the abstention rate on the natural data $\mathcal{D}_{\text{nat}}$ is 0% but the robust accuracy is also roughly 0%. Increasing $\sigma$ (for small $\sigma$) gives us higher robust accuracy for the same abstention rate. Too large $\sigma$ may also lead to degraded performance.

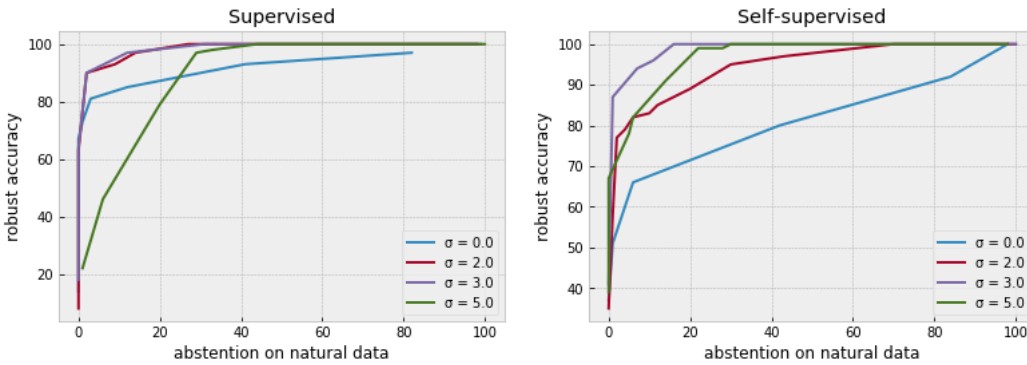

Figure 5: Adversarial accuracy (i.e., rate of adversary failure) vs. abstention rate as threshold $\tau$ varies for $n_3 = 1$ and different outlier removal thresholds $\sigma$.

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
