# OpenReview forum: "On the Power of Abstention and Data-Driven Decision Making for Adversarial Robustness"
_ICLR.cc/2021/Conference — Reject_

### Official Review · AnonReviewer1 · 2020-10-22
**clarity issues, unclear whether the setting fits within the literature**

**Rating:** 3
**Confidence:** 2

**Review:**

Summary: The authors propose a connection between abstention and robustness to adversarial examples. Specifically, the authors contend that without the ability to abstain, any classifier can be fooled by an adversarial perturbation in the feature space. They additionally provide results and experiments concerning the proper selection of a hyperparameter that tunes the abstention.

Pros: The authors include many theoretical analyses. A good effort is made to address the problem from several relevant angles.

Cons:

The paper is very difficult to read.
 - The introduction does not provide a clean line of thought motivating the paper. I am not aware that the attack model considered, that of an adversary allowed to make arbitrarily large moves in a subset of feature space, with no constraints on the input space, exists elsewhere in the literature. The work cited (Brown et al. 2018) requires that unrestricted adversarial examples remain unambiguous to human judges.

 - The abstract cites results for "any classifier" when in fact the result seems to be for a nearest-neighbor style classifier, which is unusual given that the setting is deep networks.

- The variables used throughout are difficult to keep track of.

Major issues:

- Theorem 4.1: This is obviously not true for all classifiers. The proof is very difficult to follow, and there are no useful details given in Figure 1. The result seems like it may be true by virtue of the fact that KNN with K=1 defines a hyperplane between any two points. Then a randomly chosen vector with probability 1/2 + epsilon inevitably crosses such a hyperplane eventually, but there's no reason to believe such a value in feature space could be reverse engineered or even lies within the range of F.

- The testing in Section 7.1 does not seem to include the use of non-adversarial test samples. In evaluating whether or not a threshold is too strict to be of use, it would be necessary to evaluate on in-domain test samples as well as the training set. These results seem likely to be overfit to the training data.

- The testing setup is unclear. Perhaps there are further details in the referenced papers, but it is not even clear to me how many classes are in the set. Is it only two, given the baseline of a linear classifier?

- The clarity of the paper is severely lacking. It is difficult to follow the contribution of many of the theorems, especially concerning their generality or lack thereof

Given that I find these issues too concerning to recommend publication, I have not carefully checked all of the proofs.

---

> ### Author Response · Authors · 2020-11-21
> **Response to cons and major issues:**
>
> --- We thank the reviewer for their valuable feedback. We have provided an updated introduction clarifying the motivation, and also provided informal intuition to help understand our theoretical results. We have uploaded a new version of our paper with a substantially revised introduction and other appropriate edits to reflect this, and respond to individual comments below.
>
> Cons:
>
> The paper is very difficult to read.
> The variables used throughout are difficult to keep track of.
>
> --- We apologize, and have clarified instances pointed out by reviewers in our update. We supplement all our formal results with informal intuition.
>
> The introduction does not provide a clean line of thought motivating the paper. I am not aware that the attack model considered, that of an adversary allowed to make arbitrarily large moves in a subset of feature space, with no constraints on the input space, exists elsewhere in the literature. The work cited (Brown et al. 2018) requires that unrestricted adversarial examples remain unambiguous to human judges.
>
> --- We have revised the introduction to make the motivation and related work more clear based on reviewer feedback. We agree the studied adversarial model does not exist in the literature, but several other ‘beyond norm-based’ and ‘feature-space’ models exist due to limitations of the norm-bounded models, and ours is a theoretically clean formulation.
> Our movements also remain unambiguous to human judges, as long as the feature subspace (n_3) is small enough (see paragraph 3 in updated introduction).
>
> The abstract cites results for "any classifier" when in fact the result seems to be for a nearest-neighbor style classifier, which is unusual given that the setting is deep networks.
>
> --- The lower bound indeed applies to *any classifier* (implemented on the feature space) as claimed in the abstract, while the upper bound is proved for abstention-equipped nearest-neighbor style classification of feature embeddings which may be obtained (say) using deep networks.
>
> Major issues:
>
> Theorem 4.1: This is obviously not true for all classifiers. The proof is very difficult to follow, and there are no useful details given in Figure 1. The result seems like it may be true by virtue of the fact that KNN with K=1 defines a hyperplane between any two points. Then a randomly chosen vector with probability 1/2 + epsilon inevitably crosses such a hyperplane eventually, but there's no reason to believe such a value in feature space could be reverse engineered or even lies within the range of F.
>
> --- Figure 1 (now Figure 2) is referenced by section 3.2. We apologize if it seems related to Theorem 4.1. We have fixed this and supplied a new figure (Figure 3) and an example scenario to explain the intuition for this theorem.
> Our result indeed applies to all classifiers. Our attack model assumes that the adversary can directly modify features, or in other words make (potentially unbounded norm) changes to inputs which can be modeled as movements within a small subspace in the feature space (see related work on feature-space attacks). Note that this is an abstraction to capture a class of perturbations that the adversary can potentially perform, there is no reverse engineering or inversion of F mapping involved (as is typical for feature-space attacks).
>
> The testing in Section 7.1 does not seem to include the use of non-adversarial test samples. In evaluating whether or not a threshold is too strict to be of use, it would be necessary to evaluate on in-domain test samples as well as the training set. These results seem likely to be overfit to the training data.
>
> --- The natural error rate (E_nat) reported in our table is error rate on non-adversarial (in-domain) test samples.
>
> The testing setup is unclear. Perhaps there are further details in the referenced papers, but it is not even clear to me how many classes are in the set. Is it only two, given the baseline of a linear classifier?
>
> --- CIFAR-10 (used for Table 1) is a well-known benchmark with 10 classes. Note that the linear protocol baseline refers to the standard practice of using a linear decision boundary on deep network embeddings, and can be used for multi-class classification.
>
> The clarity of the paper is severely lacking. It is difficult to follow the contribution of many of the theorems, especially concerning their generality or lack thereof
>
> --- We apologize, and have clarified instances pointed out by reviewers in our update. We supplement all our theorems in the main text with a discussion of their applicability and implications, and separately elaborate on the assumptions.

---

### Official Review · AnonReviewer4 · 2020-10-28
**A theoretical paper with fundamental results; uncertain practical import**

**Rating:** 6
**Confidence:** 3

**Review:**


This paper proves some fundamental facts about classifiers that can't abstain (provide a non-classification) and their robustness to adversarial perturbations. In Sec. 4, they provide a result that such classifiers are always vulnerable to adversarial perturbations in a technical sense. In particular, there will always be a class in which most training examples can be randomly perturbed in a way that an incorrect label will result nearly half the time. In Sec 5, they propose a modified nearest-neighbor classification algorithm, with two parameters that control abstention and "noise removal". They provide upper bounds on error in a random subspace attack scheme, and refine/loosen these results in several more specific/general scenarios. In Secs. 6 & 7, they discuss methods to tune the two parameters and provide experimental evidence of their theoretical results.

#### Strengths & Weaknesses

First, the strengths: I found the paper to be well-organized, and mostly well-written. It aims to tackle a pretty fundamental problem, and provides some clear and simple results in relation to a simple algorithm. For the statements I checked, the paper was technically sound.

As for the weaknesses, I found some of the mathematical exposition to be hard to follow. The proofs and sketches could really benefit from some diagrams and perhaps some simple example scenarios. Additionally, I found myself unsure of the practical gains from their algorithm. They make a single comparison to a linear classifier, and it's almost certain that the comparison would not be as favorable against any method that allows for classification in a non-binary fashion (with some level of confidence, and thus some natural level of abstention).

#### Recommendation

Based on the above, I gave a rating of 7. I did not give a higher score, as I am not sure of the practical relevance of the suggested algorithm. On the other hand, the theoretical results seem solid and are of a pretty fundamental nature. I do provide this score with a grain of salt, as I was not able to check all the theoretical statements, which are the backbone of this paper, in my opinion.

UPDATE: I am downgrading my score to a 6. Based on the opinions of my fellow reviewers, it seems that perhaps the theoretical results are based on scenarios that are too simplistic for the community at hand. Moreover, there are clearly some readability issues, based upon the reactions of the other reviewers.

#### Clarification Questions & Suggestions

1. As suggested, perhaps it would be helpful to discuss or consider existing methods that provide classification with some level of confidence. It seems that these automatically provide some notion of abstention when confidence is not high enough. I found it surprising that none of these were mentioned.
2. I noticed that the supplementary material contains a few references on examples of the subspace threat model. I feel like a sentence in the main text would be helpful to provide some context for the relevance of this threat model.
3. In the proof of Thm. 4.1, the existence of a label \\(y^*\\) with volume fraction less than 0.5 exists by virtue of having at least two labels. Even with an abstention option, existence would hold, I believe.

---

> ### Author Response · Authors · 2020-11-21
> **Response to weaknesses and clarification questions/suggestions:**
>
> --- We thank the reviewer for their valuable feedback. We clarify why abstention based on existing typical (ell_p-norm based) confidence estimation techniques do not work in our setting, and elaborate further on the practical motivation for our model and results in our revised introduction. Response to individual comments follows.
>
> The proofs and sketches could really benefit from some diagrams and perhaps some simple example scenarios.
>
> --- We thank the reviewer for the suggestion and have added more diagrams in the updated version of the paper, as well as example scenarios e.g. for Theorem 4.1.
>
> Additionally, I found myself unsure of the practical gains from their algorithm. They make a single comparison to a linear classifier, and it's almost certain that the comparison would not be as favorable against any method that allows for classification in a non-binary fashion (with some level of confidence, and thus some natural level of abstention).
>
> --- The comparison to the linear classifier provides a baseline for natural error. Notice that most typical classifiers used on the ‘feature space’ have linear decision boundaries, and confidence is a function of the distance from these boundaries. Therefore abstention bands are also hyperplanes which may be crossed by the perturbation vector (in any direction) for sufficiently large perturbations (for a proof, note that for any vector the two extremities will be high confidence and differently labeled). This implies that, except for 100% abstention, the robust error for these models will be 100% (same as with no abstention). Our algorithms, in contrast, exhibit a nice abstention-accuracy trade-off.
>
>
> Clarification Questions & Suggestions
> As suggested, perhaps it would be helpful to discuss or consider existing methods that provide classification with some level of confidence. It seems that these automatically provide some notion of abstention when confidence is not high enough. I found it surprising that none of these were mentioned.
>
> --- Existing methods typically use linear decision boundaries on the feature embeddings. As explained above, abstention is useless here, we add this discussion to our experiments. We remark that such abstention might be useful in other settings though (e.g. ell_p norm-based).
>
> I noticed that the supplementary material contains a few references on examples of the subspace threat model. I feel like a sentence in the main text would be helpful to provide some context for the relevance of this threat model.
>
> --- Thanks for the suggestion, we fix this in our update.
>
> In the proof of Thm. 4.1, the existence of a label y* with volume fraction less than 0.5 exists by virtue of having at least two labels. Even with an abstention option, existence would hold, I believe.
>
> --- Yes, but the rest of the bounded feature space considered is not necessarily labeled as some other class, so the adversary might not succeed if we properly use abstention.

---

### Official Review · AnonReviewer2 · 2020-10-29
**Important problem, but the theoretical formulation seems flawed and (consequently) experiments are not quite meaningful either .**

**Rating:** 4
**Confidence:** 4

**Review:**

Summary:

This paper studies, through a provable approach, whether abstaining (i.e., refusing to answer) can be beneficial for achieving small adversarial/robust error in settings where the input is potentially adversarially perturbed. The paper proves a separation between the power of models with and without abstain. In particular, it is shown that for a certain adversarial model (more about this below) when we force the model to answer without an abstain option, it will have high adversarial error, but when abstain is allowed, it can have small adversarial error as well as small abstention rate in certain settings. The paper then studies algorithms for robust contrastive learning in which they map the inputs into high-dimensional spaces and then aim to classify them using an abstain-enabled model based on 1-NN. The paper studies ways to adjust the parameters of the model as the data comes in an online fashion (divided into batches). They show how to achieve sublinear regret in such settings. They then compare linear classifiers with their own (1-NN style) classifiers and show advantages in robustness with such models when abstaining is allowed.

Pros:

Not many previous works have studied the role of abstention in adversarial attacks (aka adversarial examples / evasion attacks). This work is the first to aim for a provable separation. This is a very natural and potentially impactful direction. The ideas for the algorithm design (through a data driven approach) could lead to useful methods that have practical values.

Cons:

I think the theoretical separation is not that meaningful due to two issues:
1. The robustness is defined for an un-natural perturbation model: it is a mixture of random and adversarial (i.e., the perturbations are allowed to be in a *randomly selected* subspace) but that is not the main issue. The main issue is that the amount of perturbation in the subspace is *unbounded*. This means the adversary can basically perturb the point to an arbitrarily far point where the *ground truth* also changes. Therefore, it is not cleat at all if the perturbed point would indeed be mis-classified or not, which seems to be the minimum requirement to call something adversarial example. Here I want to contrast the noise model with, say, ell_p-based noise model that is extensively studied in the literature to clarify the issue. The idea there is that, e.g. in the case of images, bounded ell_p perturbation preserves the ground truth (in that case human’s judgement). So, an attack that finds images with small ell_p distance with a different classified label would be misclassified. Here, nothing like that could be said as perturbations are arbitrarily long.
2. To see (a different but related) issue with the definition used for robust error assume a function f(.) completely learns the concept correctly and have zero error. Then on the one hand, such model should not be able to have an adversarial example, because any perturbation would be *correctly* answered (i.e., imagine a change in a cat image to modify it into a dog picture and when the model says it is a dog, we count this as error). However, the definition used in this paper would still allow to prove *unconditional* adversarial error for the model. Note that previous works (e.g., the cited work of Madry et al.'18) are (sometimes implicitly) defined for a setting that the perturbation cannot change the ground truth (e.g., bounded perturbation of images do not change human judgement, so if the label changes it would be misclassified) but here the noise allows arbitrarily far perturbations.

-  It seems the experiments compare the new method (with possible abstention) with a linear classifier that is not designed to be robust.
I think a fairer way to show the advantage of abstain is to show that your method (with abstain) can beat another previous method that was designed to be robust (e.g., using traditional adversarial training). That would show a real jump in what we can do with abstention.

Due to the above reasons, I think the theoretical and the experimental contributions can be interpreted in a limited way, and hence I am more inclined to recommend rejection.

Main comments:

In Algorithm 1: line 2: do you do this in some order? e.g., if two points are at distance less than sigma, you remove one of them or both of them?

Discussions after Theorem 5.1 somehow interpret it as showing some form of (inherent) trade-off between success probability and abstention rate on *normal* (not adversarial) inputs. But that does not seem to be necessarily the case. For example, going back to the case of images. Note that the input distribution (e.g., images in CIFAR-10) keep their concept label even after perturbation (e.g., human judgement). Now, one can either ask a robust model to output a label *even when images are perturbed* or be allowed to abstain when a perturbed image is given. In the latter case, a model can actually have 100% accuracy on the normal inputs, while it might have a lot of abstain on adversarially perturbed points. The disparity between my example and the message of Theorem 5.1 seems to be either stemming from the fact that you allow arbitrarily long perturbations (that will eventually change the label) or that 1-NN based approaches are not sufficiently powerful here.

Assumption 1 page 5:
"We assume that at least 1 − delta fraction of mass of the marginal distribution D_{F(X)|y} over..."
Is this for every y?
Also, can you discuss whether Assumption 1 typically correct on real data?

In your experiments (reported in Table 1 (page 8)) how much the numbers change if you aim to get an adversarially perturbed point misclassified (by further restricting what constitutes as a legitimate adversarial example). My objection above to the theoretical formulation and proofs does not prevent you from (potentially) showing a separation in these experiments by really forcing the adversarial examples to be misclassified.

Is your approximate adversary provably approximating the robust error?

###############################

Further comments (and typos):

The label y appears (twice) in the proof of page 4 as non-math (missing $..$).

D_nat does not seem to be the best choice to represent the abstention rate on normal data (at least it is hard to guess it based on the notation).


***** post rebuttal comment *****

Thanks for sharing the response. Unfortunately, the very basic issue with the definition used in this paper, and its implications to practice, remains unsolved.

To clarify the definition, you just need to focus on the following simple example: what if the model has zero risk/error?  If you perturb a point, it would still be correctly classified. Yet, they still show that adversarial examples are inevitable even in this setting. This already shows something is fundamentally wrong with the definition used.

Your response is that the attacker/adversary will not get to change the label, but only the features. But please note: the adversary *is not allowed* to choose the label. The adversary picks the features, and it is up to the model to correctly classify it or not. If an attacker changes the picture of a cat to to a picture of a dog, the neural net (or any other model) should call this dog (and if does still calls it cat it would be a a mistake not the other way around). The ground truth (i.e., the concept function) determines what is correct and what is not.

The above issue is not imaginary, it has real affect on the experiments, and as I said, it is important to report in the experiments whether or not they attacks lead to *actual misclassification*.

I hope these comments will help improving the paper, since as I said, the topic of this paper is a very important one, and so exactly because of this, it is important to have the basics right.

---

> ### Author Response · Authors · 2020-11-21
> **Response to main comments:**
>
>
> In Algorithm 1: line 2: do you do this in some order? e.g., if two points are at distance less than sigma, you remove one of them or both of them?
>
> --- Both points are removed by the algorithm. We clarify this in the updated text. Typically only one of them might be an outlier, but removing both gives reasonable theoretical and empirical results.
>
> Discussions after Theorem 5.1 somehow interpret it as showing some form of (inherent) trade-off between success probability and abstention rate on normal (not adversarial) inputs. But that does not seem to be necessarily the case. For example, going back to the case of images. Note that the input distribution (e.g., images in CIFAR-10) keep their concept label even after perturbation (e.g., human judgement). Now, one can either ask a robust model to output a label even when images are perturbed or be allowed to abstain when a perturbed image is given. In the latter case, a model can actually have 100% accuracy on the normal inputs, while it might have a lot of abstain on adversarially perturbed points. The disparity between my example and the message of Theorem 5.1 seems to be either stemming from the fact that you allow arbitrarily long perturbations (that will eventually change the label) or that 1-NN based approaches are not sufficiently powerful here.
>
> --- We are describing a trade-off between success probability on perturbed data (not natural data, as in your example) and abstention rate on natural data (as a tune the amount of abstention). We cannot get 100% robust accuracy if we never abstain in our model, and as our experiments show that the robust accuracy increases when we increase the abstention rate.
>
> Assumption 1 page 5: "We assume that at least 1 − delta fraction of mass of the marginal distribution D_{F(X)|y} over..." Is this for every y? Also, can you discuss whether Assumption 1 typically correct on real data?
>
> --- Yes, it is assumed for every label class. It is a fairly standard generative assumption that simply states that one can cover most of the distribution of a given label class with balls of a fixed radius, each having a small lower bound on the density contained. It holds for small values of delta,beta if the class data is reasonably clustered and we have sufficient amount of sampling (or equivalently, sufficiently large radius \tau), which is typical for real feature embeddings (to which our result applies). We add this discussion to our update.
> For the CIFAR-10 dataset, a trivial covering with a single ball (N=1) already gives that our assumption holds for delta<0.05 and beta<0.01 for both self-supervised (for all radii > 8.0) and supervised (for all radii > 3.5). Cleverer class-specific coverings give stronger results.
>
> In your experiments (reported in Table 1 (page 8)) how much the numbers change if you aim to get an adversarially perturbed point misclassified (by further restricting what constitutes as a legitimate adversarial example). My objection above to the theoretical formulation and proofs does not prevent you from (potentially) showing a separation in these experiments by really forcing the adversarial examples to be misclassified.
>
> --- We cannot think of an easy way to compute this, as this would require the knowledge of labels for the perturbed points, but there is no simple way to invert the feature embedding. We remark that changing a few features or in a small subspace (n_3 = 1 in our experiments) is unlikely to alter human perception, so the labels should remain unaffected.
>
> Is your approximate adversary provably approximating the robust error?
>
> --- Our approximate adversary (Algorithm 5 in the Appendix) provides empirically close estimates for small threshold \tau, but does not enjoy any worst-case approximation guarantees. It is however useful only for large n_3, where our exact algorithm (Algorithm 4) is not very efficient, which is not the most relevant setting for our model and only included for completeness.

---

> ### Author Response · Authors · 2020-11-21
> **Response to cons in experiments:**
>
> It seems the experiments compare the new method (with possible abstention) with a linear classifier that is not designed to be robust. I think a fairer way to show the advantage of abstain is to show that your method (with abstain) can beat another previous method that was designed to be robust (e.g., using traditional adversarial training). That would show a real jump in what we can do with abstention.
>
> --- Standard adversarial training approaches update the network parameters by minimizing adversarial loss but still use a classifier with linear decision boundary on the network embedding and classify the entire space (without the ability to abstain). Our experiments also use a linear decision boundary on the network embedding (and not just a linear classifier) and compare it with a non-parametric nearest-neighbor based boundary with varying abstention rates. With the linear boundary and no ability to abstain, the robust error is 100% in our model as we can always cut through the decision boundaries even if the adversary is restricted to a few directions (compare e.g. with Shamir et al. 2019 where they alter a few pixels). In fact, this is provably the case (100% robust error) even with distance-based abstention for linear boundaries (except when abstention is 100%). We choose to include the non-adversarial training because the lower natural error provides a stronger baseline for natural error. \\
> To summarize, methods designed to be robust in norm-based models fail completely here. Intuitively, this is because they confidently classify points far away from the decision boundary, even though there are no training examples to justify this, and tweaking a single feature can break them. When appropriately equipped with abstention, they will also likely exhibit abstention-accuracy trade-offs under their own adversary models.

---

> ### Author Response · Authors · 2020-11-21
> **Response to cons in theory and model definition:**
>
> --- We thank the reviewer for their useful feedback which helped us clarify the model definition (e.g. stressing the feature-space aspect) and improve the motivation. We have uploaded a new version of our paper with a substantially revised introduction and several other edits to clarify our results and their significance, and respond to individual comments below.
>
>
> "I think the theoretical separation is not that meaningful due to two issues:
>
> The robustness is defined for an un-natural perturbation model: it is a mixture of random and adversarial (i.e., the perturbations are allowed to be in a randomly selected subspace) but that is not the main issue. The main issue is that the amount of perturbation in the subspace is unbounded. This means the adversary can basically perturb the point to an arbitrarily far point where the ground truth also changes. Therefore, it is not cleat at all if the perturbed point would indeed be mis-classified or not, which seems to be the minimum requirement to call something adversarial example. Here I want to contrast the noise model with, say, ell_p-based noise model that is extensively studied in the literature to clarify the issue. The idea there is that, e.g. in the case of images, bounded ell_p perturbation preserves the ground truth (in that case human’s judgement). So, an attack that finds images with small ell_p distance with a different classified label would be misclassified. Here, nothing like that could be said as perturbations are arbitrarily long."
>
> ---We would like to emphasize that our adversarial model is a model for what adversarial perturbations look like in *feature space*.  In particular, if the network is not Lipschitz, then movement within an ell_p ball in the input space could produce quite large movements in feature space. Ideally, researchers will find ways to make useful Lipschitz deep networks, but until then we wish to understand what can be done if we acknowledge the adversary may be able to move points by a large distance in feature space.  Note that these large changes may not change ground-truth, because they really relate to the effect of the network architecture on a small change to the input. \\
> Naturalness: Our model definition is a natural way to bound directionality of the adversary. It is equivalent to an average (uniform or biased) of the adversaries with different directional capabilities. A simple motivation is a defense application where an adversary might hack into and completely control a small random subset of features, and abstention may be viewed as detecting adversarial influence. Changing a small subset of a large number of features (or linear combination thereof) does not alter human judgment (see also, related works on feature-space attacks). \\
> Some limitations of norm-bounded attack: Several classes of perturbations, e.g. changing a single pixel or altering a stylistic feature, can break the norm bounds without altering human judgment on the class label.
>
>
> "To see (a different but related) issue with the definition used for robust error, assume a function f(.) completely learns the concept correctly and have zero error. Then on the one hand, such model should not be able to have an adversarial example, because any perturbation would be correctly answered (i.e., imagine a change in a cat image to modify it into a dog picture and when the model says it is a dog, we count this as error). However, the definition used in this paper would still allow to prove unconditional adversarial error for the model. Note that previous works (e.g., the cited work of Madry et al.'18) are (sometimes implicitly) defined for a setting that the perturbation cannot change the ground truth (e.g., bounded perturbation of images do not change human judgement, so if the label changes it would be misclassified) but here the noise allows arbitrarily far perturbations."
>
> --- As discussed above, arbitrarily far perturbations are considered in the feature space, so the adversary can change the features but not the labels. Like several other works, our assumption that the adversary does not change the ground truth is implicit in the sense that changing a few features would not change the ground truth, we clarify this in the updated introduction.

---

### Official Review · AnonReviewer3 · 2020-11-09
**Writing needs to be improved. Theorems statements are not clear and they seem not to be relevant to adversarial robustness research.**

**Rating:** 4
**Confidence:** 4

**Review:**

This paper studies the power of abstention in robust classification. A classifier that has the power of abstention can refuse to answer the query because it is unsure of the answer. For robust classification, the abstention power enables the classifier to refuse adversarial queries, if the query is detected as an adversarial example.

The paper first shows a negative result on the possibility of robust classification. After reading the statement of theorem and the proof multiple times, it is not clear what the theorem is stating. The theorem only says that the adversary can flip the label using arbitrary large perturbations, which seems to be a trivial statement. In the comments bellow I have listed several questions to get a clear understanding of this theorem.

After the negative result, the paper studies the effect of abstention by showing a positive result on the 1-nearest neighbor classifier. The idea is simple, whenever the query is far from its nearest neighbor in the training set, the classifier refuses to answer. This clearly provides some lower bound on the robustness as long as the data is well separated. This result cannot be used for actual image datasets that are used in practice because 1NN will definitely not have good accuracy even without abstention. Also, the images are not well-separated at all. However, the authors still run some experiments by considering the adversarial attacks in the feature space (the noise is added in the feature space instead of input space.). They show that using some good feature representation, they can get acceptable robust accuracy using their method. However, it is important to note that this will not have any meaningful effect on the real datasets classification tasks.

As for evaluation, I find the idea of paper in studying the provable effect of abstention exciting. However, the theorems that are proved lack clarity and significance. I suggest the authors to re-write the theorem in a more understandable way with all parameters clearly explained. From a technical point of view, it seems that the theorems that are proved are not really relevant to adversarial perturbations. The definitions of adversarial perturbations seem arbitrary and not aligned with standard definitions.

Comments/Questions to Authors:

Theorem 4.1:
1- The current statement of theorem is trivial. You don't provide any bound on the size of perturbation which makes the theorem not very useful.
2-The statement of theorem says a random vector v. What does that mean? Are you considering robustness to random noise or adversarial noise? If so, how is this related to adversarial examples?
3-In the proof it says R=\frac{r_\delta \sqrt{n_2}}{\delta} is large enough to provide some property about the balls of size r_\delta. Doesn't this statement require some distributional assumptions for data distribution?
4- There are already some negative results about adversarial robustness that the paper could refer to.

Theorem 5.1:

1-The definition of \Epsilon^x_adv is not clear at all which makes the whole theorem not really understandable.
2-It sounds to me that the bound on the error could be well beyond 1? If not, you should explain why. what if n3=n2?
3-  What does this sentence mean? " The adversary is allowed to corrupt F(x) with arbitrarily large perturbation in a uniform-distributed subspace S of dimension n3". Again, what do you mean by uniform distributed?
4-In general, defining adversarial perturbation in a subspace with smaller dimension seems not standard. Did you choose this type of perturbation for a specific reason or just for the sake of proving the theorem?
5- The proof of this theorem does not sound to be rigorous. Are you assuming that without any perturbation, the 1NN classifier is 100% accurate?

---

> ### Author Response · Authors · 2020-11-21
> **Response to comments in reviewer's summary of our paper:**
>
> We thank the reviewer for their useful feedback which helped us explain the model definition and the motivation for our model better. We have uploaded a new version of our paper with a substantially revised introduction and several edits to clarify our results and their significance, and respond to individual comments below.
>
> “The theorem (4.1) only says that the adversary can flip the label using arbitrary large perturbations, which seems to be a trivial statement.”
>
> --- Recall that our model allows the adversary to perturb the feature vector of each example by an arbitrary amount *but only in a random direction*.  The motivation is that most deep network models are not Lipschitz: small adversarial perturbations to the input can lead to large perturbations in feature space.  But this does not necessarily mean the adversary can make these large perturbations in feature space in any direction it wishes.  Our goal is to study a specific model of this form to understand what it would take to make such a network robust. What we show is that the ability to abstain is crucial.
> To answer your question, the adversary cannot always flip the label with arbitrarily large perturbations in our model, since it cannot make the perturbations in arbitrary directions. We have a probabilistic result which claims there is at least one class where the adversary will succeed with significant probability when we are forced to classify, no matter what surface we use as the decision boundary, or how well-separated the classes are in the feature space.
>
> “This result cannot be used for actual image datasets that are used in practice because 1NN will definitely not have good accuracy even without abstention. Also, the images are not well-separated at all.”
>
> --- Note that the abstention-equipped 1NN algorithm is applied as a classifier on the feature space, but not the input space, i.e., on top of deep network features (since our attack is defined in the feature space). Our result implies that if you replace the linear classifier in the last layer of neural networks with an abstention-equipped 1NN, you can provably bound the robust error (in our attack model).
> The images are well-separated in good feature embeddings like via contrastive learning. We only require F(x) to be well-separated and not x, in our assumptions. \\
> As our experiments in Section 7 show, this indeed gives us good natural as well as robust (under our model) accuracy for image datasets. \\
> We also remark that standard linear-boundary classifiers, even when equipped with a confidence-based abstention (points too close to decision boundary are abstained) lead to provably 100% robust error under our feature attacks (see discussion in Section 7). So the insight that nearest-neighbor style classification and abstention can provide good defense for feature-space attacks is in itself interesting.
>
>
> “However, it is important to note that this will not have any meaningful effect on the real datasets classification tasks.”
>
> --- We expect our work to have twofold impact --- 1) we lay the theoretical foundation for a new clean (beyond-Lp-norm) adversarial model where results have implications for real feature-space attacks, and perhaps more significantly, 2) we establish a provable trade-off in using abstention for defense in an adversarial setting which may shed a light for other defense models.
>
> “ However, the theorems that are proved lack clarity and significance.”
>
> --- We have worked on improving the clarity based on the valuable feedback. We believe, among other things, we provide a first formal abstention-based gap, and a first provable optimization for the induced trade-off in an adversarial defense setting.
>
> “The definitions of adversarial perturbations seem arbitrary and not aligned with standard definitions.”
>
> --- The definition of our model differs from the norm-bounded attack based definitions (see limitations below). We have revised our introduction section to emphasize how it fits with related beyond-Lp-norm models, and included other feature-space attacks for reference. \\
> Limitation of norm-bounded attack: Several classes of perturbations, e.g. altering a stylistic feature, can break the norm bounds without altering human judgment on the class label.

---

> > ### Author Response · Authors · 2020-11-21
> > **Response to questions/comments for Theorem 4.1:**
> >
> > 1- The current statement of theorem is trivial. You don't provide any bound on the size of perturbation which makes the theorem not very useful.
> >
> > --- As explained above, our model bounds the directionality of the attack, and we evaluate the probability that a direction-restricted adversary may succeed. The probabilistic argument is not immediately obvious, and this theorem is important in formally establishing that the ability to abstain is crucial in our model. We add further intuition and an example scenario in our update.
> >
> > 2-The statement of theorem says a random vector v. What does that mean? Are you considering robustness to random noise or adversarial noise? If so, how is this related to adversarial examples?
> >
> > --- Only the direction of the adversarial perturbation is random, and the adversary is free to adversarially choose the magnitude of v, so it is not a random noise. In contrast, common norm-bounded models bound the magnitude and allow all directions (albeit in the input space). See also: discussion on model definition above. \\
> > Relation to adversarial examples: Just like changing a few pixels in a large image can force a model to misclassify (Shamir et al. 2019) without changing the human judgment on its class, changing a few of many features (or linear combinations thereof) would preserve ground truth but might induce misclassification. The randomness aspect may simply be viewed as averaging the effect of attacks over all directions (since restricting to a fixed set of directions is quite arbitrary).
> >
> > 3-In the proof it says R=\frac{r_\delta \sqrt{n_2}}{\delta} is large enough to provide some property about the balls of size r_\delta. Doesn't this statement require some distributional assumptions for data distribution?
> >
> > --- No. r_\delta is fixed in the previous line, and is data-dependent but finite for any \delta. The statement about R is a purely geometric claim that holds for any r_\delta and \delta --- any two balls of radius R in n_2 dimensions, if their centers are at most r_\delta apart, have a large (1-\delta) overlap in their volumes.
> >
> > 4- There are already some negative results about adversarial robustness that the paper could refer to.
> >
> > --- Thanks for the suggestion. We refer to negative results of (Shamir et al. 2019) which talk about large perturbations but in the input space. For feature-space attacks, we are only aware of empirical negative results. We included these in the introduction and related works, but now also summarize them in context in Section 4 in our updated version.

---

> > ### Author Response · Authors · 2020-11-21
> > **Response to questions/comments for Theorem 5.1:**
> >
> > 1-The definition of \Epsilon^x_adv is not clear at all which makes the whole theorem not really understandable.
> >
> > --- The definition simply formalizes the intuition that the adversary is allowed to corrupt feature mapping of x with arbitrarily large perturbation in a uniform-distributed subspace S of dimension n_3. That is, consider all subspaces of dimension n_3 of the feature space. One subspace is chosen uniformly at random, and the adversary can now perturb a feature vector to anywhere in this subspace. This is a natural generalization of perturbing in a uniformly random direction. This is explained in detail in Section 3.1 when our model is defined. \\
> > In context of the theorem 5.1, \Epsilon^x_adv is the robust error (in our attack model) when classifying x. Since our bound on  \Epsilon^x_adv is independent of x, averaging this over the distribution of x gives the same bound for expected robust error over test distribution as well.
> >
> > 2-It sounds to me that the bound on the error could be well beyond 1? If not, you should explain why. what if n3=n2?
> >
> > --- Yes, the error bound could be greater than 1 (and vacuous) when n_3-n_2 is too small, but this would correspond to a powerful adversary which can perturb by arbitrary amounts and in “most” directions and likely impossible to obtain low error against. It is more interesting to consider the case where n_2 is large and n_3 is fixed or small relative to n_2
> > (otherwise the true label of the perturbed point may change). \\
> > If n_3=n_2, the adversary will always win as it has access to the entire space, and error happens with probability 1 for any (non-abstain) prediction.
> >
> > 3- What does this sentence mean? " The adversary is allowed to corrupt F(x) with arbitrarily large perturbation in a uniform-distributed subspace S of dimension n3". Again, what do you mean by uniform distributed?
> >
> > --- Consider the set of all subspaces of the feature space (assumed to be n_2-dimensional Euclidean) which have dimension n_3. By saying “uniform-distributed”, we mean a subspace S is drawn uniformly at random from this set.
> >
> > 4-In general, defining adversarial perturbation in a subspace with smaller dimension seems not standard. Did you choose this type of perturbation for a specific reason or just for the sake of proving the theorem?
> >
> > --- See model discussion above. We view this model as a reasonable alternative to norm-bounded models which have known limitations. We limit the geometry instead of the magnitude of perturbations. A simple motivation is a defense application where an adversary might hack into and completely control a small random subset of features, and abstention may be viewed as a detection of adversarial influence.
> > See also: related work in updated introduction.
> >
> > 5- The proof of this theorem does not sound to be rigorous. Are you assuming that without any perturbation, the 1NN classifier is 100% accurate?
> >
> > --- No, we are not assuming without any perturbation the 1NN classifier is 100% accurate. The theorem upper bounds the robust error, which by definition includes the cases where the adversary can succeed. If the example is already misclassified by the 1NN, the adversary will succeed with zero perturbation unless we successfully abstain on it, which happens for the selected tau (always for Theorem 5.1, and with high probability for the stronger Theorem E.1).

---

### Author Response · Authors · 2020-11-21
**Paper update overview**

We would like to show our sincere gratitude for the valuable feedback that we derived from all reviewers. We took all the comments very seriously and regarded them as great opportunities to enhance the overall quality of our submission. We have tried to address all of their concerns and revised our paper accordingly. We will also answer directly to each review comment.

We summarize the most notable changes to our submission in our updated version:
- Emphasized that our model is a *feature space* attack model in the abstract and introduction.
- Substantially revised the introduction section to clarify the model motivation and to accommodate other reviewer feedbacks.
- Added a brief discussion for the intuition of Theorem 4.1, with a simple example scenario.
- Added more informal intuition, literature context and example scenarios throughout for our theoretical results.


We sincerely hope that this revision effort meets your expectations. We are happy to provide further clarification if any of the reviewers’ concerns are not answered.

---

### Decision · Program_Chairs · 2021-01-07
**Final Decision**

**Decision:**

Reject

**Comment:**

The paper considers the problem of abstention in robust classification. A number of issues were identified in the formal framework and the writing was also not up to scratch. The authors should take into regard the very many constructive suggestions made by the reviewers in preparing a revision.